# The Importance of School Leaders’ Attitudes and Health Literacy to the Implementation of a Health-Promoting Schools Approach

**DOI:** 10.3390/ijerph192214829

**Published:** 2022-11-11

**Authors:** Simona Betschart, Anita Sandmeier, Guri Skedsmo, Tina Hascher, Orkan Okan, Kevin Dadaczynski

**Affiliations:** 1Institute for Research on Professions and Professional Learning, Schwyz University of Teacher Education, 6410 Goldau, Switzerland; 2Department of Teacher Education and School Research, University of Oslo, 0317 Oslo, Norway; 3Department of Research in School and Instruction, Institute of Educational Science, University of Bern, 3012 Bern, Switzerland; 4Department of Sport and Health Sciences, Technical University of Munich, 80333 Munich, Germany; 5Department of Health Science, Fulda University of Applied Sciences, 36037 Fulda, Germany; 6Zentrum für Angewandte Gesundheitswissenschaften, Leuphana Universität Lueneburg, 21335 Lueneburg, Germany

**Keywords:** health-promoting school, health literacy, school leaders, attitudes

## Abstract

Schools are considered as important settings for health education and the promotion of functioning, healthy people. Health-promoting schools (HPS) take a holistic approach to health promotion that addresses all school levels and members of the school community. Although it is assumed that school leaders are crucial in the implementation of HPS, little is known so far about which specific factors are relevant. The purpose of this study was to analyze the role of school leaders’ attitudes, health literacy, and health status in HPS implementation. An online survey was conducted in June 2021, and the final sample consisted of N = 358 school leaders. Regression and moderation analyses were conducted to answer the research questions. The results showed the relevance of health literacy and attitudes toward HPS for the implementation of HPS. By contrast, only some aspects of health status are relevant. Attitudes toward HPS moderate the effect of health literacy on HPS implementation, with positive attitudes toward HPS amplifying the effect. Implications for practice and future research are discussed.

## 1. Introduction

Health is considered a fundamental human right and is essential to our social and economic development [1]. During the COVID-19 pandemic, awareness of public health promotion increased; more than ever, the development and maintenance of physical and mental health were of interest to the worldwide community [2]. Health promotion and research on the topic have, therefore, gained increased public attention, despite being important long before the pandemic. In the Ottawa Charter for Health Promotion, health was declared a fundamental human right, and its promotion was announced as a general objective [3]. The Charter stresses the value of implementing health promotion in settings where people spend a large part of their lives. Schools have consequently become one of the most relevant places for health promotion for several reasons. First, this is because a broad population of children and adolescents can be reached through schools, representing essentially the only socialization setting where all children and adolescents can be reached [4]. Schools also offer windows of opportunity for intervention, as young people can be reached at an early age, even before health-related attitudes and behaviors have been established. Early investments in health education could hinder social inequalities and foster the integration of functioning, healthy people into our society [5]. Second, scholars increasingly agree that mental and physical health is related to students’ academic performance and achievements and school outcomes [6,7,8]. Poor health conditions among students or teachers affect the quality of the learning and teaching processes, and vice versa [7,9,10,11,12]. In fact, the whole school environment has a considerable influence on mental and physical health. For example, unfavorable school conditions, such as a poor school climate, low performance expectations, or a lack of classroom structures and rules, can have a negative impact on the health of students and school staff (e.g., [12,13,14]). Even though health is not the primary goal of most schools, schools and health promotion ultimately share the same goal: to foster healthy, educated individuals who can make successful societal contributions [8].

The fact that schools represent an ideal setting for health promotion has led to the generation of numerous health-promotion programs [5,15,16]. In view of the countless possibilities, the question arises as to how health-promotion programs can best be designed. The World Health Organization (WHO) explicitly advocates for a holistic approach to school health promotion [17], as it accounts for all components of a school community [18]. To enable as many schools as possible to implement a holistic health-promotion approach, the concept of Health-Promoting Schools (HPS) was developed. Compared to more shorter-term, isolated health interventions, this approach considers all school levels and addresses all school community members [19]. In other words, HPS seek to promote health through the school curriculum (formal), the ethos or environment (informal values, attitudes, and physical environment of a school), and families and communities [8]. The Schools for Health in Europe Network (SHE) supports European schools in implementing HPS. The SHE recommends including the following six core aspects: (1) healthy school policies, (2) the school’s physical environment, (3) the school’s social environment, (4) individual health skills and action competencies, (5) community links, and (6) health services [19]. HPS aim to achieve profound and lasting development across the entire school community regarding health promotion, and this can be achieved when schools themselves take the initiative and implement interventions tailored to their specific needs and conditions [20]. However, the implementation and maintenance of HPS are complex, as different actors, contents, and levels must be considered. Therefore, leadership and the coordination of such processes are essential [21,22,23,24]. Studies show that school leaders act as gatekeepers for health promotion and, thus, decisively influence schools’ prevention and health-promotion activities [25,26]. School leaders not only initiate health-promoting activities, but also continuously accompany the health-promotion actions of their schools by handling motivational, organizational, and content-related aspects, and integrating health promotion into daily practice [13,24]. The significance of school leaders in health promotion is also identified by Zumbrunn et al. and Gieske and Harzd [16,27] who ascribe a central role to school leaders in ensuring and coordinating health-promotion activities in their schools. Although school leaders are repeatedly credited with an important role in health promotion, little is known about how they direct health promotion or which characteristics are important for their associated leadership practices [23,25]. Most of the previous research has concentrated on school leaders’ knowledge and attitudes, as well as the roles and responsibilities of school leaders in the initiation, support, and maintenance of HPS. Other relevant aspects, such as school leaders’ concrete strategies or their internal resources (e.g., health status), have hardly been explored thus far [25]. Although it is reasonable to assume that the health of school leaders is central to their activities and, in particular, their engagement with HPS, few empirical studies have elucidated this relationship [28]. 

### 1.1. Key Concepts and Previous Research

The WHO defines health as, “A state of complete physical, mental and social well-being and not merely the absence of disease or infirmity” [17]. This definition is based on Antonovsky’s salutogenetic concept [29,30], which implies that health is not merely about the absence of disease, but rather that it has to be seen as a continuum from “healthy” to “ill,” on which we constantly move back and forth over the course of life [29,31]. Where we are on the continuum depends on several factors, which can be both externally (e.g., stressors and viruses) and internally influenced (e.g., coping skills and resources). Antonovsky emphasized that individuals who have sufficient resources to cope with the daily demands placed on them generally remain healthier [29,31]. His line of argumentation can also be found in the transactional stress theory, which is based on the assumption that stress occurs when a situation is perceived as threatening and when the demands of the environment exceed the coping abilities [32,33]. To survey the health status of individuals, it is, therefore, necessary to focus on two aspects: (1) the current state of well-being and (2) the current state of stress, emerging from the degree to which individuals perceive situations in their lives as uncontrollable, unpredictable, and overloaded (perceived helplessness [PH]) relative to their subjective feeling of control (perceived self-efficacy [PSE]). A 2009 Swiss study found that three-quarters of school leaders reported high to very high levels of well-being, while about 30% reported difficulties recovering after work [34]. A study from Germany [28] showed that 24% of school leaders surveyed reported low well-being and 12% reported very low well-being. There is additional evidence that school leaders belong to a particularly stressed occupational group [28,35]. Work-related stress is known to lead to decreased job satisfaction, depression, and burnout [35,36,37]. Especially since the COVID-19 pandemic, school leaders have experienced an extraordinary amount of pressure, sleepless nights, and limited resources for decision-making [38]. Swiss school leaders reported increased workhours, a straining workload, and increased stress in the first year of the pandemic [39]. Sormunen et al. [40] argued that this increase in demand was a risk factor for health problems and for potential negative effects on school quality and effectiveness. Although there is no empirical evidence yet, it is reasonable to consider school leaders’ health status as a key factor influencing HPS in general and during the pandemic in particular. There are two possible directions of the effect that must be considered. On the one hand, it is possible that a low health status among school leaders means that they have a reduced capacity for further innovation, such as health promotion. On the other hand, it is also conceivable that a low health status makes school leaders conscious of the issue and leads them to prioritize school health promotion.

In contrast to school leaders’ health, several studies have investigated school leaders’ attitudes toward HPS, showing that their mindsets are highly relevant to the implementation of health promotion [16,21,24]. For example, Clarke et al. [21] found that US school leaders perceived the relationship between children’s health and their learning as a major driver in promoting school health. Zumbrunn et al. [16] concluded that shared awareness of health issues and shared attitudes toward health promotion among Swiss school leaders and teachers are essential for implementing HPS. Similarly, a Taiwanese study found that school leaders’ motivation to maintain HPS was significantly related to higher levels of HPS implementation and sustainability [41]. When school leaders recognize the importance of health, it can lead to far-reaching changes throughout the school community [23,26]. However, this awareness requires basic knowledge about health and its far-reaching significance. The ability to identify and manage health-related information to maintain and promote health can be subsumed under the concept of health literacy [42,43], which encompasses much more than just knowledge of or the ability to acquire and understand health information. The concept describes the ability to care adequately for one’s own health and the health of others [42]. This includes, for example, changing one’s life circumstances and behaviors such that a healthy lifestyle becomes possible [42,44]. During the pandemic, health literacy played a particularly important role due to the amount of health-related information distributed to school leaders [45]. In Germany, for example, a study showed that limited health literacy among male school leaders was associated with low levels of health-promoting school activities [46].

### 1.2. Key Features of the Swiss Contexts and Background of the Study

As mentioned earlier, previous studies have shown that school leaders play a vital role in HPS implementation. How this role is designed in detail (e.g., level of autonomy and available resources) depends highly on educational governance structures in the education system. For this reason, it is crucial to describe how the situation during the pandemic was for school leaders in the Swiss contexts. Switzerland is characterized by a federal structure, similar to Germany, and the country is divided into 26 states (cantons), all of which have a high degree of decision-latitude in most governmental matters. For the educational landscape of Switzerland, this federal structure means that considerable differences exist between the individual cantons regarding the school system. As a result, school leaders’ tasks and responsibilities differ depending on the canton in which they work. This setup leads, among other things, to health promotion being approached and implemented differently. The range of health-promotion programs in Switzerland is enormously diverse. Schools have a great deal of freedom when it comes to choosing, implementing, and financing programs [16]. A nationwide survey on the implementation of health promotion and prevention makes clear how diverse the programs used are. Each school features a distinctive activity profile that adapts to the present problems, problem-awareness, and resources [16]. However, there are hardly any systematic data on the extent to which health promotion is anchored in the training of teachers and school leaders. Initial training and professional development of teachers and school leaders are organized through the universities of teacher education, which do not have uniform curricula. Despite these differences, there are some aspects that are similar for all cantons. In most curricula, health is integrated as an interdisciplinary topic (e.g., in physical education, home economics, or general education). In addition to being anchored in the curriculum, health is promoted and health-promotion activities are coordinated by various networks and NGOs (e.g., Gesundheitsförderung Schweiz, Radix, Schulnetz21, or bildung und gesundheit Netzwerk Schweiz) [47]. These organizations pursue various goals. For example, they strive for a stronger legal anchoring of school health promotion and strengthening health promotion in teacher training. They also support schools in implementing various health-promotion programs.

During the pandemic, there was only a relatively brief period from March to May 2020, during which national guidelines were applied and schools were closed nationwide. During the rest of the pandemic and since then, only recommendations have been issued nationally, and concrete decisions have largely been left to the cantons. The cantonal authorities, in turn, delegated much of the decision-making power to individual schools, which resulted in an extremely prominent level of decision-making authority being given to school leaders. On the one hand, this offered many opportunities for local adaptation, but on the other hand, it was perceived as a burden [48]. School leaders had to make sense of a large amount of diverse and sometimes conflicting information provided by authorities and handle diverse expectations from teachers, students, and parents.

### 1.3. Research Questions and Hypotheses

In summary, school leaders are considered important to HPS implementation, especially in times of pandemics. This article examines the importance of school leaders’ health literacy in HPS implementation, beyond the impact of their own health status and attitudes toward health promotion. The research questions and the hypotheses for this study are based on identified gaps in the existing research literature.

Three research questions guided this study: RQ1: What is the relationship between school leaders’ health status (well-being, PH, and PSE) and attitudes toward HPS and the level of HPS implementation?RQ2: Does school leaders’ health literacy explain additional variances in the level of HPS implementation when accounting for other factors (well-being, PH, PSE, and attitudes)?RQ3: Is the relationship between health literacy and HPS implementation moderated by other school leaders’ characteristics (well-being, PH, and PSE)?

The following hypotheses guide the analysis, based on the literature review above: 

**H1.1:** 
*School leaders with poor well-being report lower levels of HPS implementation in their schools.*


**H1.2:** 
*School leaders with high PH report lower levels of HPS implementation in their schools.*


**H1.3:** 
*School leaders with high PSE report higher levels of HPS implementation in their schools.*


**H1.4:** 
*School leaders with positive attitudes toward HPS report higher levels of HPS implementation in their schools.*


**H2:** 
*School leaders’ health literacy explains additional variances, even when accounting for health status and their attitudes toward HPS.*


For the interaction effects in RQ3, no hypotheses were formulated, as an explorative analysis approach was chosen.

## 2. Materials and Methods

This study is part of the project Health Promotion in Schools and Coping with the COVID-19 Crisis (HEPISCO), which is associated with the COVID Health Literacy (COVID-HL) network (https://covid-hl.eu/ (accessed on 30 October 2022)), which is an open science and research community fostering inquiry in the fields of health literacy, health information, and digital health [49]. Many of the instruments used were adaptations (e.g., for the school- or COVID-19-specific context) of standardized scales that are also used by other network members. 

### 2.1. Study Design and Study Population

Using Unipark, an online survey was conducted among school leaders all over Switzerland. A link to the survey was distributed through the newsletters of the Swiss associations for school leaders in the German-, French-, and Italian-speaking parts of Switzerland (Verband Schulleiterinnen und Schulleiter, VSLCH; Conférence latine des chefs d’établissements de la scolarité obligatoire, CLACESO), through the newsletter of the Schwyz University of Teacher Education and promoted through social media. Before distributing the survey, the questionnaire was piloted with members of the Association for School Leaders. Data were collected during June 2021, constituting a convenience sample. An extensive data cleanup was, therefore, necessary after completion to ensure that the final sample comprised only school leaders (i.e., speeders and individuals with incomplete information were excluded). The target group was defined as school leaders and members of the school leadership team. In total, 162 respondents from the original sample were excluded. The adjusted sample consisted of N = 358 school leaders. Table 1 shows the characteristics of the sample. As this was a convenience sample, the sample cannot be described as representative. A comparison with data from the Swiss Federal Statistical Office [50] shows that school leaders from primary schools are underrepresented. This is also reflected in the gender distribution. More male than female school leaders participated, although, according to the official statistics, there are more female school leaders. This difference is likely because male school leaders are more represented in secondary and vocational schools. However, the age distribution can be considered representative [50].

### 2.2. Measures

As the study was conducted while the pandemic and its effects were still of practical relevance, some of the scales were adapted to ask school leaders specifically about their experiences and assessments during the pandemic.

The main outcome variable in this study was the level of HPS implementation [49]. School leaders were asked to what extent HPS activities were conducted at their schools during the COVID-19 pandemic (e.g., “At our school, students learn how to get enough exercise despite the limitations due to the coronavirus”). The scale builds on existing previous work [20] and maps the three core dimensions of holistic approaches to school health promotion (curriculum, school environment, and extracurricular cooperation/cooperation). The original scale consists of 15 items with a four-point response scale with regard to their agreement (1 = does not apply at all, 2 = rather does not apply, 3 = rather applies, 4 = completely applies). Based on a principal component analysis (PCA) for all three language versions of the survey, we excluded two items due to low communalities and factor loadings (“At our school, students are involved in the planning of prevention and health promotion measures” and “At our school, [digital] spaces for social interaction and exchange are created despite the corona-related restrictions”). The internal consistency of the final scale of 13 items is good, with a Cronbach’s alpha of 0.85.

School leaders’ health status, attitudes toward health promotion, and health literacy were included as explanatory variables. The measurement of the health status was based on two components, well-being and stress. The current state of well-being was measured using the WHO Well-Being Index (WHO-5) [51,52]. Respondents rated five statements (e.g., “Over the last two weeks I have felt cheerful and in good spirits”) on a Likert scale from 1 (at no time) to 6 (all of the time). The total score is multiplied by four, and the final score ranges from 0 to 100, with 0 representing the worst imaginable well-being and 100 representing the best imaginable well-being [52]. Internal consistency is good (α = 0.87) To assess the stress experienced by school leaders during the pandemic, we used an adapted version of the German Perceived Stress Scale (PSS, [53]), which was adjusted specifically for the school (e.g., “How often have you felt that things at your work at school were going your way?”) and COVID-19 contexts (e.g., “The questions in this scale are about your feelings and thoughts regarding your work during the COVID-19 pandemic in last month”). The scale assesses the degree to which individuals perceive situations in their lives as uncontrollable, unpredictable, and overloaded relative to their subjective coping abilities, represented by the two subscales PH and PSE [53]. The PH subscale consists of six items with good internal consistency (α = 0.84), while the PSE scale includes four items with a questionable internal consistency (α = 0.66).

Attitudes toward health promotion were measured using a scale developed by Dadaczynski et al. [46]. School leaders were asked to what extent they agreed with six different statements (e.g., “The health of my staff and my students is very important to me”) on a 5-point Likert scale (1 = agree completely, 5 = do not agree at all). Internal consistency is (α = 0.67). As the scale is still used in the early stages of research, this can be considered as acceptable [54].

Health literacy was measured using a COVID-19-specific scale developed by Okan et al. [55], which was assessed based on the European Health Literacy Survey Questionnaire (HLS-EU-Q), which measures participants’ perceived difficulty or ease when dealing with health information [44]. The scale consists of 16 questions (e.g., “On a scale from very easy to very difficult, how easy would you say it is to find information about the coronavirus on the internet?”) that participants rank on a 4-point scale (1 = very easy, 4 = very difficult). Internal consistency is high (α = 0.91).

### 2.3. Statistical Analyses

For statistical analyses, SPSS version 28.0 was used. For the preliminary analysis, box plot diagrams were created to detect any outliers. Only a few outliers became visible, of which two extreme ones were excluded. 

For a data overview, univariate analyses, such as mean values and standard deviations, were calculated. Subsequently, Pearson’s correlations were conducted to analyze the correlation structures among the variables. To answer the research questions, multiple linear regression analyses were conducted using the Enter method. Predictors were entered as follows: (model 1) school leaders’ health status and attitudes toward HPS; (model 2) school leaders’ health status, attitudes toward HPS, and health literacy; (model 3) school leaders’ health status, attitudes toward HPS, health literacy, and two-way interactions between health literacy and each of the school-leader-related antecedents (well-being, PH, SE, attitudes, and health literacy). For this model, all continuously measured variables were mean-centered, and interaction terms were calculated by obtaining the product of these centered variables. 

In addition, we performed a simple slope analysis and plotted the moderation effect using PROCESS for SPSS [56], a convenient procedure for conducting a simple slope analysis that aids the interpretation of the interaction effect [56].

In all models, the sociodemographic characteristics of the school leaders (age and gender) and the school (type and size of the school and proportion of students with low socioeconomic status (SES)) were controlled.

## 3. Results

The level of HPS implementation is comparable to a recent study conducted in Germany (*M* = 2.96, *SD* = 0.48) [46]. On the WHO-5 scale, school leaders reported low well-being (*M* = 60.76, *SD* = 19.12). School leaders rated their PSE as 3.75 on average (*M* = 3.75, *SD* = 0.52) and their PH as 2.96 (*M* = 2.96, *SD* = 0.74). The scale ranged from 1 to 5. Attitudes toward health promotion were rated as 4.68 on average (*M* = 4.68, *SD* = 0.31). 

### 3.1. Correlations 

Correlations show that most of the variables of interest are related to the dependent variable (HPS implementation) in the expected direction (see Table 2). School leaders’ well-being (*r* = 0.15) and PSE (*r* = 0.20) are positively related to the implementation of HPS. On the other hand, PH is negatively correlated (*r* = −0.11). School leaders’ attitudes toward HPS are positively related to HPS implementation (*r* = 0.17), and the same can be said of school leaders’ health literacy (*r* = 0.27). The factors are all related in the expected direction, and the effect sizes can be classified as weak to moderate [32]. In addition, it becomes apparent that well-being, self-efficacy, and helplessness are strongly correlated.

### 3.2. Regression Analysis

To analyze the relationship between the explanatory (independent) variables and HPS implementation, a regression analysis was conducted using a stepwise approach (see Table 3). In model 1, the school leaders’ age and the proportion of students with a low SES at their school constitute significant predictors of the level of HPS implementation. Older school leaders tend to report higher levels of HPS (*b* = 0.20, *t*(352) = 3.72, *p* < 0.001). In addition, schools with a higher proportion of students with a low SES report significantly lower levels of HPS implementation (*b* = −0.12, *t*(352) = −2.32, *p* < 0.05). Further, PSE (*b* = 0.19, *t*(348) = 2.90, *p* < 0.01) and attitudes toward health promotion (*b* = 0.16, *t*(348) = 3.18, *p* < 0.01) constitute relevant predictors of the level of HPS implementation. For school leaders’ general well-being and PH, no significant associations were found. Together, these factors explain 11% of the variance in HPS implementation, *R*^2^ = 0.11 (*F*(9, 348) = 5.92, *p* < 0.001). School leaders’ health status was operationalized using the WHO-5 and the PSS consisting of PSE and PH. While no significant results could be found regarding general well-being (WHO-5), mixed results emerged from the PSS. The PSE subscale constitutes a relevant predictor, while the PH subscale does not. 

In model 2, the addition of health literacy leads to a significant increase in the explained variance, *R*^2^ = 0.144 (*F*(10, 347) = 7.0, *p* < 0.001; Δ*R*2 = 0.034, *p* < 0.001). School leaders with high health literacy reported significantly higher levels of HPS implementation (*b* = 0.20, *t*(347) = 3.83, *p* < 0.001). 

In model 3, interaction effects were also included. The main effects of health literacy and attitudes toward HPS remain significant. In addition, the interaction effect of health literacy*attitudes toward HPS becomes significant (*b* = 0.13, *t*(343) = 2.57, *p* < 0.05; Δ*R*2 = 0.011, *p* = 0.077). For other interactions, no significant effects were found.

To analyze this interaction in detail, simple slope analyses were conducted, as described in the section below. The analysis shows that attitudes toward HP amplify the positive relationship between HL and HPS under certain conditions: HL is significantly associated with HPS under the condition of positive attitudes toward HP (*β* = 0.32, SE = 0.07, *p* < 0.001) or average attitudes toward HP (*β* = 0.25, SE = 0.06, *p* < 0.001). Under the condition of poor attitudes (*β* = 0.06, SE = 0.08, ns), HL has no significant effect on HPS. Figure 1 graphically represents the simple slope analysis.

## 4. Discussion

HPS are considered as the best way to promote health among students, educators, and non-teaching staff. When implementing HPS, research has shown that school leaders play a defining role and act as gatekeepers [5,23,26]. It is, therefore, paramount to understand better which characteristics of school leaders are related to the level of HPS implementation. Several factors, such as school leaders’ health status, attitudes toward HPS, or health literacy, are considered to play a role in HPS implementation in schools [13,21,46,57]. Especially during the pandemic, health literacy became a key competency for school leaders [43,45], as there was a vast amount of information they had to process and put into practice. 

The aim of this article was to understand the role of school leaders’ health literacy in school health promotion beyond the other factors, such as school leaders’ own health status and attitudes toward health promotion.

We expected that school leaders with a poor health status would report lower levels of HPS implementation at their schools (Hypotheses 1.1–1.3). We measured the health status of school leaders using two components: well-being and stress. School leaders reported low well-being, compared to the average population (*M* = 60.76, *SD* = 19.12). According to a study by Eurofound, the average value in the normal population is 66 for women and 68 for men in Austria and 62 for women and 67 for men in Germany [58]. Values below 50 indicate a reduced sense of well-being [52]. Unfortunately, data from Switzerland are not available. In our data, well-being had no significant relation to HPS, and only one dimension of stress—PSE—was related to HPS implementation, which means that only Hypothesis 1.3 can be confirmed, and Hypotheses 1.1 and 1.2 have to be rejected. A classification of these mixed findings is difficult, because there is limited research on the influence of school leaders’ health on HPS implementation [13]. For this reason, we discuss these results, including empirical and conceptual considerations. The WHO-5 assesses well-being by asking about an individual’s health status over the preceding 14 days [52]. Well-being as measured in this study can, therefore, be considered a proximal health outcome. It is possible that more distal health parameters could be more strongly associated with HPS implementation. In the future, research should focus on different health measures to examine the relationship between school leaders’ health and HPS implementation. Such studies could focus on stress and specific symptoms of exhaustion. Several studies have shown that school leaders suffer from high mental stress and report psychological complaints significantly more often than other occupational groups [35,59,60,61]. A first attempt in this direction was made by applying the PSS, which measures PH and PE in overwhelming situations. The latter showed a significant relation to HPS, confirming the results of several studies that underline the importance of school leaders’ self-efficacy for many outcomes [62]. Self-efficacy, as a person’s belief in their ability to succeed in a particular situation, is an important predictor of an individual’s behaviors and performances [63,64]. It has been shown that the self-efficacy of school leaders is related to their job satisfaction [65,66], students’ academic achievements, or the collective performance of the entire school [62,67,68]. Furthermore, principals with high self-efficacy are more likely to perceive their school as a learning organization and to be more confident that implementing changes is possible [69]. However, further research is needed on the specific relationship between school leaders’ self-efficacy and HPS implementation.

The attitudes of school leaders are comparable to those of a recent study in Germany (*M* = 4.68, *SD* = 0.31) [46]. In line with Hypothesis 1.4, we found that school leaders with positive attitudes toward HPS reported higher levels of HPS implementation at their schools. Similar results can also be found in several other studies (e.g., [7,11,48]).

In summary, the answers to RQ1 reflect the current state of research: The impact of the health status at the school leader level remains unclear, whereas school leaders’ attitudes toward HPS are undeniably of great importance. 

Based on our analysis, the answers to RQ2 are much clearer. According to Okan et al. [55], the health literacy reported by school leaders in this study can be classified as sufficient [55]. In line with Hypothesis 2, we confirmed that health literacy plays a significant role in HPS implementation, even when accounting for other factors. These findings are consistent with previous studies [45,46,57]. The current study extends these findings by asking whether the effect of health literacy on HPS is moderated by other factors (RQ3). The results show that the influence of health literacy is largely dependent on attitudes toward HPS. Thus, high health literacy hardly has any effect on HPS implementation among school leaders with negative attitudes toward HPS. Only with medium and positive attitudes toward HPS does the effect of health literacy become significant. In turn, this means the effect of high health literacy on HPS implementation can be strengthened with developing positive attitudes toward HPS. Notably, the above argumentation assumes that health literacy precedes the implementation of health promotion. However, as this study is a cross-sectional study, no causal conclusions can be drawn. It is also possible that school leaders’ health literacy is promoted by their active implementation of HPS. In that case, health literacy would not be an antecedent but rather a result from school health promotion. Further longitudinal studies could shed more light into the causal relationship between these two factors.

### 4.1. Practical Implications

The outbreak of the COVID-19 pandemic has led to a situation in which health has become an issue that requires every school to put it on its agenda [70]. This increased awareness could now be used to promote HPS in the education system, as well as in the overall society. 

Our findings demonstrate how crucial attitudes toward health promotion are for the implementation of HPS. These attitudes should be fostered by integrating the topic of health and health promotion into professional development programs for school leaders, as suggested by Dadaczynski et al. [46]. In Switzerland, such programs are offered by universities of teacher education, of which there are currently 14. Each has its own curricula for the training of school leaders. In addition, school leadership training has a very recent history in Switzerland, as many cantons first established the formal role and function of school leaders shortly after the turn of the millennium. Before that, schools were run primarily by teachers in the role of primus inter pares [71] but answering to a municipal governing body [72]. An overview of the curricular content of school leadership programs shows that health promotion is only included as a topic to a small degree [73]. In addition to the finding of the importance of health literacy, our study shows the importance of health literacy in implementing whole-school health promotion. As suggested by Dadaczynski et al. [46], it would be desirable to integrate health literacy in school leaders’ education. In addition, the findings of our study show, in addition, that health literacy without positive attitudes toward HPS is not very effective. Consequently, it is not enough simply to promote health literacy among school leaders in their training. As suggested above, a focus on promoting positive attitudes toward HPS needs to be included in school leaders’ professional training. This request is also made by Deschesnes et al. [74], who imply that school leaders should be educated about the HPS approach and its benefits for their schools. This line of argumentation is adopted by Clarke et al. [21], who found that school leaders often feel a lack of necessary expertise and capacity when it comes to the HPS implementation and are, therefore, unwilling or too fearful to dedicate more time to health-promoting activities. Attitudes indeed become more important when school leaders find themselves in highly stressful situations, such as a pandemic. Kruse and Huber [75] found that school leaders often reduce their activities to operational day-to-day business and tend to disregard “additional” tasks, such as staff and school quality development or health promotion. Thus, if health promotion is not understood as a fundamental element of a school, it is quickly pushed aside as an optional accessory. This topic takes on additional importance in times of teacher shortages and in times of crisis when emergency responses are needed. Studies have made clear that the attitudes and health literacy of school leaders can have a decisive influence on the health, job satisfaction, and stress experiences of teachers [27,41]. In terms of an HPS approach, the promotion of attitudes toward HPS and health literacy in school leaders is, therefore, about not only student health but also teacher health and the retention of teachers. 

The professional practice of school leaders is dependent on professional development offers, governmental structures, and available resources. Therefore, a clear position of the cantonal authorities and municipalities is fundamental to prioritizing further holistic school health promotion, for example, in terms of developing concept papers or guidelines. This would support school leaders and help clarify the importance of HPS and its benefits. Moreover, sufficient time and resources are needed to be made available for HPS to be given a higher priority by the entire school community [26,76].

### 4.2. Limitations and Suggestions for Future Research

The present study has some limitations. First, the data consisted of a convenience sample, which makes generalizability difficult. We cannot discard the possibility that the participating school leaders have a special interest in health and school health promotion. Thus, the survey should be replicated with a representative sample. Second, an online survey was conducted; therefore, people with limited access to digital devices or with limited technical skills could have been excluded. Still, this risk can be considered low, as most school leaders have their own computers in Switzerland. Especially during the pandemic, the necessity of good technical equipment became more pronounced. Third, the data were of a cross-sectional nature, which disallows any causal conclusions. Thus, the effects of school leaders on HPS implementation should be analyzed in studies with a longitudinal design. Fourth, all results were based on self-reports, which is particularly problematic for measuring HPS implementation, as school leaders may answer in a socially desirable manner or overestimate their current practices. Future studies could include a more comprehensive HPS assessment, e.g., through site visits, document analyses, or surveys of teachers, students, and/or parents.

## 5. Conclusions

The present study has shown that the PSE, attitudes toward HPS, and health literacy of school leaders were of crucial importance to HPS implementation during the pandemic. In principle, it is assumed that these findings can be transferred to the school context without a pandemic. In previous studies, both health literacy and attitudes toward HPS were found to be individually important to HPS implementation [21,46]. In this study, it also became clear that, along with health literacy, attitudes toward HPS remain a crucial predictor of HPS implementation. Therefore, it is important to promote both factors in professional development programs for school leaders and to provide support and sufficient resources for schools to implement complex and elaborate HPS interventions.

## Figures and Tables

**Figure 1 ijerph-19-14829-f001:**
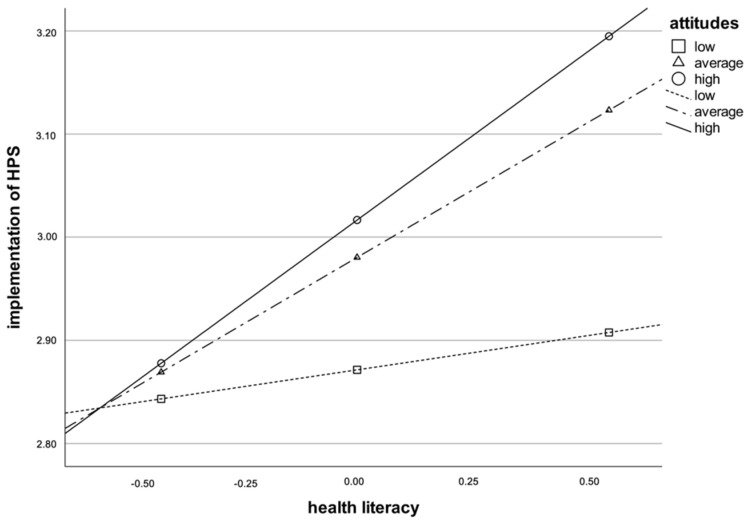
Simple slope analysis. Dependent variable: HPS implementation.

**Table 1 ijerph-19-14829-t001:** Characteristics of study participants.

Item	Category	Frequency (*n*)	Percentage (%)
Gender	Male	196	54.7
	Female	162	45.3
Age	≤45 years	96	26.8
	46 to 60 years	220	64.8
	>60 years	42	8.4
Type of school	Primary school	137	38.3
	Secondary school	67	18.7
	Comprehensive school	69	19.3
	Vocational school	75	20.9
	Special education school	10	2.8
School size (Number of students)	≤300 students	107	29.9
	301–600 students	108	30.2
	>600 students	143	39.9
Proportion of students with low socioeconomic status	≤10%	75	20.9
	11–50%	231	70.7
	>50%	52	8.4

**Table 2 ijerph-19-14829-t002:** Pearson Correlations.

	1	2	3	4	5
1. WHO	-				
2. PSE	0.54 **	-			
3. PH	−0.59 **	−0.61 **	-		
4. ATTI	−0.04	−0.02	0.02	-	
5. HL	0.17 **	0.20 **	−0.16 **	0.10	-
6. HPS	0.15 **	0.20 **	−0.11 *	0.17 **	0.27 **

Note. WHO = General well-being, PSE = Perceived self-efficacy, PH = Perceived helplessness, ATTI = Personal attitudes toward HPS, HL = Health literacy, HPS = Health promotion schools. * *p* < 0.05, ** *p* < 0.01.

**Table 3 ijerph-19-14829-t003:** Results of Regression Analysis. Dependent Variable: HPS Implementation.

Predictor	Model 1	Model 2	Model 3
Constant (SE)	2.394 (0.18)	2.406 (0.17)	2.398 (0.17)
Gender	0.096	0.083	0.088
Age	0.180 ***	0.172 ***	0.167 **
School size	−0.070	−0.051	−0.042
School level	0.097	0.088	0.087
Low SES	−0.117 *	−0.102 *	−0.094
PSE	0.192 **	0.164 *	0.166 *
PH	0.057	0.063	0.065
Well-being	0.065	0.050	0.055
Attitudes	0.160 **	0.140 **	0.144 **
HL		0.195 ***	0.197 ***
HL*PSE			0.079
HL*PH			0.035
HL*well-being			0.037
HL*attitudes			0.127 *
R^2^ corrected	0.110	0.144	0.155
F	5.92 ***	7.00 ***	5.67 ***
(df1/df2)	(9/348)	(10/347)	(14/343)

Note. Multiple linear regression analysis (Enter method), standardized β-coefficients. * = *p* < 0.05, ** = *p* < 0.01, *** = *p* < 0.001, Two-tailed tests. SE = standard error, SES = socioeconomic status, PE = perceived self-efficacy, PH = perceived helplessness, HL = Health literacy, *R^2^* = multiple determination coefficient.

## Data Availability

The data presented in this study are available upon request from the corresponding author.

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
