# Peer review of "The Importance of School Leaders’ Attitudes and Health Literacy to the Implementation of a Health-Promoting Schools Approach"

_ijerph, 2022, doi:10.3390/ijerph192214829_

Round 1

Reviewer 1 Report

This is a highly significant study on the leadership of principals in promoting school health, and I think it is appropriate for publication in your journal with the following additions. The following points should be added to the article: "In Switzerland, there is no description of how school health or health education is included in the training of school principals. If possible, it should also be mentioned how health education is included in the curriculum for the training of all teachers, or whether it is included in the training of specific subject teacher. Then, the process of becoming a principal in Switzerland should be described and discussed, as well as whether health education is included in the examination and education process. These systems vary from country to country and are not uniform. In conclusion, since the authors mention the need for principals' leadership in the dissemination of HPS even in the ordinary time, authors should be mentioned more than leadership in emergency response is needed.

Author Response

Dear Reviewers

Thank you very much for your helpful comments.

Please see in the attachment, that we have listed your comments below chronologically along the structure of the paper, and documented in the right column how the comments were addressed in the revision. We sincerely thank them for your helpful comments. They have helped to make the paper more stringent and clear.

Sincerely,

The authors

Reviewer 2 Report

Thank you for the opportunity to review this article.

I remind you that all the suggestions provided in this review are suggestions for improvement and with the aim of promoting reflection around the analyzed object.

Title:

Reflects the content covered. It is properly worded.

Summary:

The authors make a brief introduction to the topic, but do not specify its purpose and the instruments they used.

Introduction:

The main concepts under study are presented, as well as the relationship between them. The authors use relevant references to present the topic and justify its relevance. In this section, the authors present the objective of the study clearly, as well as all the guiding questions.

Methods:

In the Study Design and Study Population section, the authors present the characteristics of the sample, however, these data are already results, so they should be presented only in the results section. The same is true for Cronbach's alpha values. The authors refer that alpha values ​​are acceptable, although there is evidence that emphasizes that only 0.7 is an acceptable value, so I suggest that they refer to authors who confirm what they say.

Results:

In this section, the authors present the results and, to a certain extent, are already discussing the results. I think you should in the next discussion section. This section should be objective and present the results of the present study, as it will be better for the reader.

Discussion:

The authors compare their results with the most current and pertinent evidence available and acknowledge the main limitations of the present study. The authors substantiate the results with mostly current and pertinent literature.

Conclusion:

The authors respond to the objective of the study, as intended in this section.

Author Response

(The authors gave the same response as above.)

Reviewer 3 Report

This manuscript addresses a topic that is very important in the field of health promoting schools: the role of school leaders in the implementation of the programme and – on top of this – their health literacy as a prerequisite of the willingness of school leaders to adopt health promotion. I was already convinced that this is an important manuscript when I have read the title and the manuscript has not disappointed.

The introduction touches on the most important background to lead to the topic of health promoting schools. It is true that school leader’s attitudes have not been examined enough; therefore, the role their health status and their health literacy have on the implementation are interesting questions.

The method section is described in sufficient detail, the results answer the research questions and the discussion adequately explains the results and puts them into context.

Some minor questions/suggestions:

-          How is school health promotion implemented in Switzerland? Is there some sort of national/regional programme/ network/ financial incentives for schools? Maybe you could give just a little context

-          Are the characteristics of the study participants similar to the characteristics of school leaders in Switzerland? I would find a comparison for the characteristics gender and type of school interesting.

-          Have you asked the school leaders how long they have/their school has been adopting a health promoting school approach? Is there a possibility that the school leaders’ health literacy is not a supporting factor but rather a result from school health promotion?

-          in line 410 the word “hast” should be “has”

Author Response

(The authors gave the same response as above.)
